# Impact of Integration of Severe Acute Malnutrition Treatment in Primary Health Care Provided by Community Health Workers in Rural Niger

**DOI:** 10.3390/nu13114067

**Published:** 2021-11-14

**Authors:** Abdias Ogobara Dougnon, Pilar Charle-Cuéllar, Fanta Toure, Abdoul Aziz Gado, Atté Sanoussi, Ramatoulaye Hamidou Lazoumar, Georges Alain Tchamba, Antonio Vargas, Noemi Lopez-Ejeda

**Affiliations:** 1Action Against Hunger, West and Central Africa Regional Office, Dakar 29621, Senegal; adougnon@wa.acfspain.org (A.O.D.); ftoure@wa.acfspain.org (F.T.); 2Action Against Hunger, 28002 Madrid, Spain; pcharle@accioncontraelhambre.org (P.C.-C.); tchalain2000@yahoo.com (G.A.T.); avargas@accioncontraelhambre.org (A.V.); 3Doctorate Program in Epidemiology and Public Health, Rey Juan Carlos University, 28933 Madrid, Spain; 4Action Against Hunger, Niamey 11491, Niger; agado@ne.acfspain.org; 5Direction de la Nutrition du Niger, Niamey 623, Niger; at_sanoussi8@yahoo.fr; 6Centre de Recherche Médicale et Sanitaire (CERMES), Niamey 10887, Niger; lramatoulaye@yahoo.fr; 7EPINUT Research Group (ref. 920325), Complutense University of Madrid, 28040 Madrid, Spain; 8Department of Biodiversity, Ecology and Evolution, Faculty of Biological Sciences, Complutense University of Madrid, Jose Antonio Novais, 12, 8th Floor, 28040 Madrid, Spain

**Keywords:** severe acute malnutrition (SAM), community health workers (CHW), integrated community case management (iCCM), mid-upper arm circumference (MUAC), coverage

## Abstract

The present study aimed to assess the effectiveness and impact on treatment coverage of integrating severe acute malnutrition (SAM) treatment at the health hut level by community health workers (CHWs). This study was a non-randomized controlled trial, including two rural communes in the health district of Mayahi: Maïreyreye (control) and Guidan Amoumoune (intervention). The control group received outpatient treatment for uncomplicated SAM from health facilities (HFs), while the intervention group received outpatient treatment for uncomplicated SAM from HFs or CHWs. A total of 2789 children aged 6–59 months with SAM without medical complications were included in the study. The proportion of cured children was 72.1% in the control group, and 77.2% in the intervention group. Treatment coverage decreased by 8.3% in the control area, while the group of CHWs was able to mitigate that drop and even increase coverage by 3%. This decentralized treatment model of acute malnutrition with CHWs allowed an increase in treatment coverage while maintaining a good quality of care. It also allowed the early inclusion of children in less severe conditions. These results may enhance the Niger Ministry of Health to review the management of SAM protocol and allow CHWs to treat acute malnutrition.

## 1. Introduction

Located in West Africa’s heart, Niger is a landlocked Sahelian country whose desert covers more than two-thirds of its surface, making it one of the largest countries in West Africa with an area of 1,267,000 km^2^. The intercensal growth rate increased from 3.3% for the period 1988–2001 to 3.9% for the period 2001–2012, with a projected population estimated at 21,942,944 in 2019 according to the General Census of Population and Housing (Recensement Général de la Population et de l’Habitat, RGPH), rural households account for 82% of all households, compared to 18% for urban households [1].

Acute malnutrition is still an urgent public health problem worldwide. In 2019, emaciation continued to threaten the lives of approximately 47 million children, including 6.9% of the world’s children under five years of age. Of those, more than two-thirds of all emaciated children under five years of age lived in Asia (69%), and more than a quarter lived in Africa (27%) [2]. Malnutrition in all its forms is an underlying cause of 3.1 million deaths among children under five years of age (45% of total deaths) and is responsible for 875,000 child deaths per year [3]. According to the results from The Cost of Hunger Study in Niger, during the 2010–2014 period, child deaths related to undernutrition accounted for 42.7% of mortality cases in children under five years of age. The annual cost of child undernutrition is estimated at 289.7 billion CFA francs, which corresponds to 7.1% of the gross domestic product because of the cumulative effects of health expenditures, school expenditures, and productivity loss labor market [4].

In West Africa, the prevalence of global acute malnutrition (GAM) is 7.5% [95% C.I. 6.5–8.6] with 1.8% [1.5–2.2] severe acute malnutrition (SAM) in 2019 [2]. In Niger, the nutritional situation that year continued to be of concern, with a GAM prevalence of 10.7% [9.5–12.0], including 2.7% [2.1–3.4] SAM. In the Maradi region, the prevalence was 11.4% [9.0–14.2] and 3.4% [2.2–5.4], respectively [5]. The prevalence of GAM estimated by the link nutrition causal analysis (Link NCA) study conducted in December 2016 in Mayahi district (Maradi region) by Action Against Hunger was 11.4% [9.4–13.8%] and the prevalence of SAM was 1.7% [1.0–2.9] [6]. These figures are above the emergency threshold defined by the World Health Organization (WHO), which establishes a high prevalence of GAM as 10% [7]; hence, malnutrition remains a public health problem in the Maradi region, including the Mayahi district.

Access to health care remains low, with 48.3% health coverage at the national level according to Niger’s 2016 health statistics yearbook and 35.7% in the Mayahi district [8], and SAM treatment coverage was estimated at 41.2% according to a survey conducted by Action against Hunger that year [9]. This assessment identified the following obstacles to health care accessibility using previous survey responses: (i) geographical accessibility problems, (ii) excessive time spent by mothers or caregivers due to high workloads, and (iii) the child has been rejected in the past.

The Niger’s community management of acute malnutrition (CMAM) policy was revised in 2016 by the Nutrition Direction of the Ministry of Health. It states that the authorized treatment providers are nurses in health centers, so CHWs are not authorized to treat SAM [10]. The integrated community case management (iCCM) approach was developed in Niger in the early 2000 s. During the first phase, treatment for diarrhea, acute respiratory infection (ARI), and malaria was provided only at the health hut level by community health workers (CHWs). The CHWs are employed by the prefecture or the communities through local contracts and receive six months of health training [11].

Several authors have identified iCCM as the opportune platform to increase SAM treatment coverage [12]. This approach of integrating malnutrition treatment among the curative services provided by CHWs has been tested in other countries such as Angola, Bangladesh, Ethiopia, India, Malawi, Mali, Pakistan, and South Sudan [13]. In parallel with this new approach to integrating nutritional treatment in iCCM programs with CHWs, which is considered par UNICEF as one of the simplified approaches, there has been increasing interest in simpler treatment protocols that facilitate malnutrition management in complex contexts with low availability of human and material resources. One of these adaptations is the use of mid-upper arm circumference (MUAC) as a single criterion for entry to treatment with an expanded cutoff point that also allows the inclusion of children with low weight-for-height [14].

In this context, the aim of the present study developed by Action Against Hunger was to assess the effectiveness and impact on treatment coverage of integrating SAM management at the health hut level by nonmedical CHWs in the Mayahi health district, Maradi region in Niger, with special attention to the anthropometric criteria for admission to treatment.

## 2. Materials and Methods

### 2.1. Study Design and Location

To assess the effectiveness of adding SAM treatment into the services delivered by CHWs, a nonrandomized controlled trial was designed, including two rural communes in the health district of Mayahi: Maïreyreye (control) and Guidan Amoumoune (intervention). The control group received outpatient treatment for uncomplicated SAM from health facilities (HFs), while the intervention group received outpatient treatment for uncomplicated SAM from HFs or CHWs. Both are rural communes located in the north of the Mayahi department, in the region of Maradi, located in the Sahel zone of Niger with similar climate conditions. Sanitary map Mayahi district can be found in the Appendix A.

Both communes were comparable in terms of availability of HFs and health huts, low admission of SAM cases, population density, and distance to the district’s hospital of Mayahi. The control group included the 4 HFs in Maïreyreye, and the intervention group included the 6 HFs plus 10 CHWs identified by the Ministry of Public Health. The CHWs were trained for four days in national protocols for the management of SAM as established by Niger’s Ministry of Public Health. This training was facilitated by three trainers, one from the central level, one from the regional level, and one from the district.

All children who attend HFs or CHWs’ huts from June 2018 to March 2019 and met the inclusion criteria were recruited for the study. The inclusion criteria were the presence of mild (+) or moderate (++) edema and/or a weight-for-height (WHZ) less than −3 z-score concerning the WHO reference and/or a MUAC less than 115 mm [15]. Cases with severe edema, medical complications, or negative appetite tests were excluded from the study and referred for inpatient treatment. According to the national protocol for the management of SAM, admitted children received 170 Kcal/kg of ready-to-use therapeutic food (RUTF) daily and were followed-up weekly until recovery was reached.

### 2.2. Socioeconomic Assessment

Prior to the beginning of the intervention, a socioeconomic survey was carried out to ensure the comparability of the two communes. Due to the geographic dispersion of the target population, a two-stage cross-sectional cluster survey was conducted in each zone. The first cluster level consisted of random selection of villages connected to the HFs existing in each commune, and the second cluster level consisted of random selection of households existing in those villages. The sample size for this socioeconomic survey was calculated automatically by the NutriSurvey.ena delta software version of July 2015 based on the population figures provided by the health district considering an average household size of 7 persons and 22.3% of children under five years. Parameters included in the calculation were an expected GAM prevalence of 12.0%, desired accuracy of 5%, cluster effect of 1.5% and a 7% of expected non-responding households. Accordingly, there were 30 selected villages in each commune with 7 households per village and a total sample size of 265 households interviewed.

The survey was conducted between 22 March and 9 April 2018. Once the intervention began in June 2018, anthropometric monitoring of children 6–59 months was carried out in these same households to determine the prevalence of GAM (MUAC < 125 mm) and SAM (MUAC < 115 mm) in the control and intervention communes.

### 2.3. Treatment Coverage Assessment

The coverage of the SAM treatment in each commune was assessed through surveys conducted at baseline (April 2018) and end-line (March 2019) applying the same standardized methodology called semi-quantitative evaluation of access and coverage (SQUEAC). The SQUEAC methodology offers a reliable direct method of assessing coverage of CMAM programs, adapted to small areas, and provide a detailed analysis of program barriers. The study was conducted in 3 stages: (1) to identify areas of low and high coverage as well as reasons for coverage failure using routine program quantitative data and qualitative data collected before the survey; (2) to confirm the location of areas of high and low coverage and the reasons for coverage failure identified in Stage 1 carrying-out small surveys to identify those children with 6 to 59 months with SAM identified by a MUAC < 115 mm and assessing if they were already under nutritional treatment; (3) to provide an estimate of overall program coverage using Bayesian techniques with an specific software developed for this methodology [16].

### 2.4. Treatment Effectiveness Assessment

A disaggregated follow-up of all children admitted to treatment in the two study communes was carried out. The anthropometric severity of the children at admission by each indicator was compared between groups. The rationale for this analysis is that adding CHWs as treatment providers outside of health centers brings treatment closer to the most remote locations and this can have a positive impact by reducing the severity with which children enter treatment.

There is currently no formal definition of what a case of extreme anthropometric severity is, so there are no internationally recommended cut-off points for WHZ or MUAC in the clinical practice guidelines of the United Nations agencies. In order to assess the anthropometric severity objectively and replicable, the median and quartile distribution of each indicator is analyzed. This approach allows comparing the proportion of cases within the most severe quartile in each study group.

For the analysis of treatment outcomes, recovery was considered no edema for 14 days and WHZ ≥ −2 z score and/or MUAC ≥ 125 mm (depending on the anthropometry criteria for admission) during two consecutive follow-up visits. The time to recovery was recorded in these children discharged as cured. The other outcomes of treatment were default (when the child was absent in two consecutive visits); nonresponse (when the edema did not disappear in 21 days or a loss or no increase in weight was registered in two consecutive visits); and medical referral (when severe medical complications appeared or there was loss of appetite).

To assess the integration of nutritional treatment with the other diseases commonly treated at the primary care level, all cases of concurrent malaria, diarrhea and ARI as other non-complicated clinical signs at admission were recorded and compared between groups.

### 2.5. Statistical Analysis

Statistical analysis was performed with SPSS v.26 software. The nonparametric Mann–Whitney test was used to compare continuous quantitative variables. The effect size was calculated to assess the magnitude of the differences through Cohen’s d coefficient using an online calculator [17]. A small effect is considered when the coefficient is between 0.2 and 0.4, an intermediate effect between 0.5 and 0.7, and a large effect beyond 0.8 [18]. For the comparison of proportions, the Chi-square test was used with the Yates correction when the number of expected cases was less than 5 in more than 20% of the cross-table cells. A calculation of the adjusted probability for the time to event was made to measure the magnitude of the differences in treatment outcomes through Cox regression, obtaining the hazard ratio values for each outcome and the survival curves for recovery. The risk ratio was calculated to measure the probability of being treated from other diseases in an integrated manner with SAM treatment.

### 2.6. Ethical Considerations

All the parents or guardians of the children included in the study signed informed consent for participation. The study was approved by the National Health Research Ethics Committee of Niger (007/201 8/CNERS), and the trial has been registered at ISRCTN with identification number ISRCTN31143316 [19].

## 3. Results

The results of the socioeconomic survey performed before the intervention are shown in Table 1. Both communes are comparable for most of the indicators analyzed. Of those indicators that differed, the intervention group appeared to have houses with better roofing and greater drinking water availability in the home. It reports distance to health centers as the main barrier to health access. In contrast, the control group reported better results regarding treatment preferences, choosing health centers in greater proportion and healers in less proportion.

The prevalence of GAM identified during the intervention period was similar in the control group (*n* = 62, 11.3% [8.7–13.9]) compared to the intervention group (*n* = 76, 12.1% [9.6–14.6], *p* = 0.670), as was the prevalence of SAM (control: *n* = 15, 2.7% [1.3–4.1] vs. intervention: *n* = 17, 2.7% [1.4–4.0], *p* = 1.000).

Regarding the effectiveness assessment, the final sample after 9 months of recruitment consisted of 2022 children in the intervention group with CHWs and 767 children in the control group without CHWs. Groups did not differ in terms of sex ratio (male: control 54.8% vs. intervention 50.7%; *p* = 0.055) or age distribution (mean age: control 16.1 ± 7.1 months vs. intervention 16.3 ± 7.5 months, *p* = 0.981; ≤24 months: control 90.4% vs. intervention 89.6%, *p* = 0.555). Within the intervention group, 39.2% of children were treated by CHWs.

Figure 1 shows the results on coverage assessments at baseline and end-line. At the beginning of the study, coverage did not differ significantly between groups, while they did differ at the end of the study. The inclusion of CHWs as providers of treatment outside of HFs allowed the coverage to increase by 3.1% in one year (*p* = 0.010). In the control group, it not only did not increase but also decreased by 8.3% during the same period (*p* = 0.098).

Anthropometric measurements at admission are shown in Table 2. Children treated in the group that included CHWs as treatment providers had a less severe anthropometric condition at admission than their peers in the control group treated exclusively in the HFs. Considering those indicators used as admission criteria (absolute MUAC and WHZ), their average values were significantly lower in the intervention group. However, the size of the effect is only relevant for the WHZ. In addition, a smaller proportion of children were in the higher severity range (MUAC < 110 mm and WHZ < −3.83). A total of 78 cases with edema were recorded (0.5% of the control group and 3.9% of the intervention, *p* < 0.001). Within the children treated in the intervention group, no significant difference was found in the cases of edema of children treated by CHWs versus those treated by health personnel (4.3% vs. 3.1%, *p* = 0.172).

Of the total number of nonedematous children included in the study, 33.9% (*n* = 776) were admitted based on the MUAC criterion, 12.3% (*n* = 282) were admitted based on the WHZ criterion, and 53.8% (*n* = 1231) were classified by both criteria simultaneously. However, this classification differs between study groups so that intervention health area that involved CHWs as treatment providers showed a significantly lower percentage of children admitted by both criteria simultaneously (*p* < 0.001) (Figure 2). Inside the intervention group, there was no difference in the proportion of children classified by both criteria between the health staff and the CHWs (47.3% vs. 48.6%, *p*= 0.856).

For the whole sample, the concordance between indicators for the classification of SAM was practically zero (Kappa = −0.073, *p* < 0.001). Cross-table analysis indicates that 18.6% (*n* = 282) of the children classified by the WHZ would not enter treatment using MUAC as the sole admission criterion. Table 3 shows a simulation of the proportion of children classified by WHZ that would be included in treatment if only MUAC were used but extending its thresholds to 118, 120, and 125 mm. In all the settings (full sample, control/intervention group and HFs/CHWs within the intervention group), increasing the MUAC threshold to as an admission criterion allows almost all children identified by the WHZ to be admitted into treatment.

Table 4A shows the treatment outcomes for each study group. The health areas that included CHWs as treatment providers (intervention group) showed a significantly higher probability of recovery than the control group and greater internal mobility (internal transfers). In contrast, the nonrespondent rate was significantly higher than that of the control group. However, analyzing this outcome inside the intervention group, the health staff doubled the proportion of non-responders compared to the CHWs (9.7% vs. 4.3%) (Table 4B). Additionally, the proportion of fatal cases (0.5% vs. 2.5%) was significantly lower in the CHW group (0.5% vs. 2.5%), and the cure ratio was almost 10% higher than that reached by the health staff in health facilities.

Figure 3 shows the comparison of the survival curves for the ‘recovery’ event resulting from the Cox regression adjusted by the admission criteria. For all the length of stay recorded, the cumulative survival was higher for the intervention group and in the CHWs.

Table 5 shows the results of the recovery time of those children discharged as cured. The median time spent until the recovery of the children in the intervention group was 13 days lower than that in the control group, with an acceptable strength of association, according to Cohen’s d coefficient. Within the intervention group, the CHWs registered a lower recovery time, which was seven days lower than that in the health centers. However, the strength of this association is weak.

Table 6 shows the proportion of severely acutely malnourished children who have also been treated in an integrated manner for other conditions. More cases of diarrhea, fever, and cough were identified in the intervention group at the time of admission. During follow-up visits, the probability of treating a case of diarrhea was 3.5 times higher and 4.3 times higher for acute respiratory infections. Analysis of the intervention group results shows that the CHWs treated more cases of respiratory infection and malaria than the health personnel but fewer cases of diarrhea.

## 4. Discussion

The study results have shown how the addition of CHWs as treatment providers outside of HFs can help significantly improve service coverage. During the study period, there was a nurses strike in the HFs of both groups. This had a negative impact on the coverage figures of the control group, where treatment was only provided in HFs. In the intervention group, coverage not only did not decrease but increased slightly by 3%.

Previous studies in Angola [20], Bangladesh [21], Burkina Faso [22], Mali [23], Ethiopia [24], and Tanzania [25], have also reported highly satisfactory results in terms of coverage with this decentralized approach through CHWs, with figures exceeding 80%. In the present study the final coverage in the intervention group was not so high (52.9%) but the addition of CHWs as treatment providers outside the HF avoided the drop of coverage due to the strike.

Children treated in the group where the CHWs managed SAM in their villages had a less severe anthropometric condition at admission than children treated exclusively at HFs. The proportion of children within the lowest quartile was significantly smaller for both indicators (MUAC: 19.4% vs. 37.4%; WHZ: 21.1% vs. 32.6%). Similar results have been reported in Mali, where children treated by CHWs registered lower anthropometric measurements at admission and fewer edema cases than those attended at formal health facilities [26]. In the present study, the proportion of children with edema did not differ between treatment providers in the intervention group.

There is robust scientific evidence demonstrating the discrepancy that exists between the different anthropometric indicators in the diagnosis of SAM when applying the criteria currently recommended by the WHO (MUAC < 115 mm vs. WHZ < −3 Z-score) [27,28,29]. Indeed, other causes independent of nutritional status may explain the difference between MUAC and WHZ in the selection of children with SAM. MUAC is closely related to muscle mass and increases with age [30]. In addition, children with long lower limbs generally have a low WHZ [31], while those with a large abdominal or chest circumference have a higher WHZ [32]. This fact may be contradictory since having long lower limbs is a good indicator of good growth quality, and these children are, therefore, in most cases healthier [33]. In the present study, the results confirm the evidence for malnourished children of Niger, with a very low concordance between indicators–lower than 8%. The results show that of the children admitted to treatment, there is a higher proportion identified by MUAC as the sole criterion compared to WHZ (34% vs. 12%). However, studies using national cross-sectional data show different results, and more children were identified by WHZ (51% vs. 24%) [34].

New approaches in acute malnutrition management propose using MUAC as the sole criterion for entry to treatment but with an upper cutoff point for SAM and moderate acute malnutrition (MAM) children. This simplified protocol could be especially useful in complex contexts of low treatment accessibility, which are the settings where CHWs can have the most significant impact as treatment providers [14,35]. The present study results show that by extending the MUAC cutoff point to 125 mm, almost all children classified as having SAM by the WHZ (99.5%) would have also been included in the treatment. This result would support the simplified criteria based on expanded MUAC in future programs with CHWs in Niger. A limitation of this analysis is that it is based on children who have already been diagnosed and included in treatment. However, a previous study using a cross-sectional community-based sample in Niger with children aged 6–24 months showed that the sensitivity and specificity of MUAC regarding WHZ were maximized using a cutoff point of 120 mm for SAM and 125 mm for MAM [36]. Further similar analyses of national cross-sectional data, including children aged 6 to 59 months, are required.

Concerning treatment effectiveness, the control group did not achieve the minimum standard of 75% of cured patients required in humanitarian response [37], while the decentralized treatment model did meet this criterion. This difference in the percentage of cured patients between the control group (72.1%) and the intervention group (77.2%) seems to be due to the inclusion of CHWs as treatment providers since, within the intervention group, the health care personnel did not reach this standard either (73.1%). In contrast, the results of the CHWs markedly exceeded this standard (83.7%). These results confirm evidence from previous studies with similar methodologies conducted in Mali, where the proportion of children cured was higher in study groups involving CHWs as treatment providers as compared with those who received standard care (94.2% vs. 88.6% and 95.9% vs. 88.7%) [23,26]. Likewise, similar operational experiences with CHWs treating SAM developed in other contexts have registered cure percentages above 80% [13]. Many of the previous studies also found a significant reduction in the proportion of dropouts, although this result was not found in the present study.

The results show that the recovery time was significantly lower in the intervention group (49 vs. 36 days), and on average the children treated by CHWs in the intervention group recovered seven days earlier than those treated at health facilities. This result could be related to the lower severity at admission as previously discussed. The length of stay registered in this study is similar to that found in other studies, with CHWs treating SAM with similar discharge outcomes: 34 days in Tanzania [25], 40 days in Pakistan [38], or 39–42 days in Mali [23,26]. Children diagnosed with SAM by both criteria and those diagnosed only by MUAC needed a longer time to recovery (45 and 42 days, respectively) than those diagnosed by edema or WHZ (35 days). A similar analysis was conducted in a community-based SAM management program in Gambia, but this study found no significant difference in recovery time due to admission criteria [39].

A previous study conducted in the Maradi region of Niger found that the most frequently occurring diseases in SAM cases with complications were gastroenteritis, respiratory tract infections, and malaria [40]. One of the added values of including the treatment of SAM among the tasks already performed by CHWs is that it enables a more integrated treatment with the rest of the deadliest diseases in childhood. In the present study, CHWs treated more malaria cases and acute respiratory infection in conjunction with acute malnutrition than health workers at HFs. The Mali study mentioned above found similar results for diarrhea (36% vs. 18%) and ARI (34.8% vs. 25.2%), as well as a higher proportion of malnourished children treated for malaria (41.7% vs. 19.8%) [26]. In this study, it is suggested that the explanation for these differences may be that HFs are subjected to a heavier workload that does not allow professionals to dedicate sufficient time to diagnose other diseases when a child is admitted for the treatment of acute malnutrition.

The study design may have led to some selection bias and caution needs to be exercised when results are extrapolated to broader contexts. The study design used a target population already seeking care in each group, it is therefore possible that the design was less susceptible to recruit difficult to reach cases. The study area was supported by an international NGO and hence, may reflect performance levels associated with well-supported interventions. Further studies assessing the long-term status of children who recovered with each treatment delivery model are needed to ascertain whether the iCCM+ approach has an effect on reducing relapse and mortality rates over time.

## 5. Conclusions

This is the first study to evaluate the impact of a decentralized SAM treatment model through CHWs in Niger. The results have shown that this new model of community-based management of acute malnutrition allowed an increase in treatment coverage while maintaining good quality. It also allowed the early inclusion of children in less severe conditions, which could be related to shorter treatment lengths of stay. Unlike the control group, the intervention group with CHWs did achieve international quality standards in the treatment of acute malnutrition. CHWs reached a higher proportion of cured cases than HFs and treated more cases of other diseases in an integrated manner. Further analysis would be needed to identify how to make the work of CHWs operational on a large scale in the country, and whether the same good results in terms of effectiveness and coverage would be achieved if CHWs working in a humanitarian context.

## Figures and Tables

**Figure 1 nutrients-13-04067-f001:**
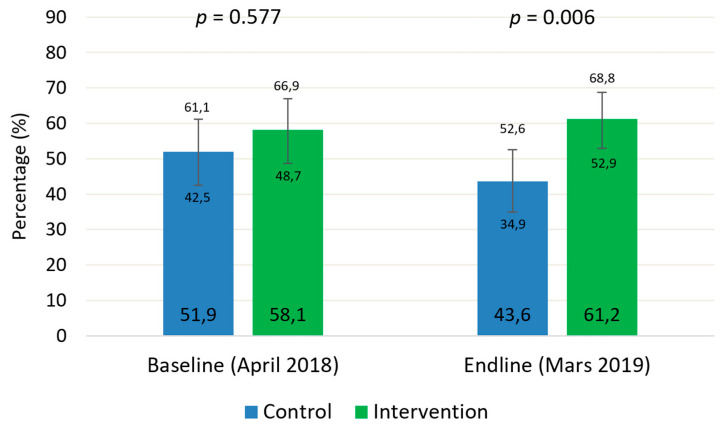
Coverage of severe acute malnutrition treatment in the study groups before and after the intervention.

**Figure 2 nutrients-13-04067-f002:**
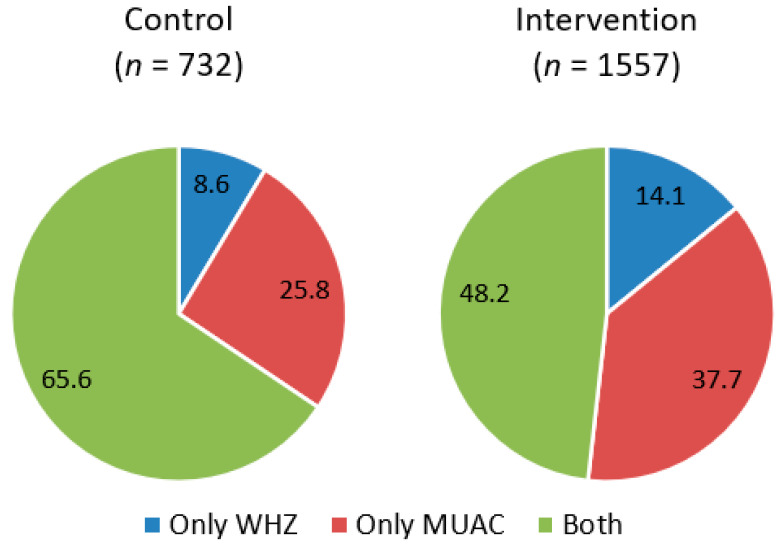
Sample distribution by anthropometric criteria for admission to treatment compared between groups.

**Figure 3 nutrients-13-04067-f003:**
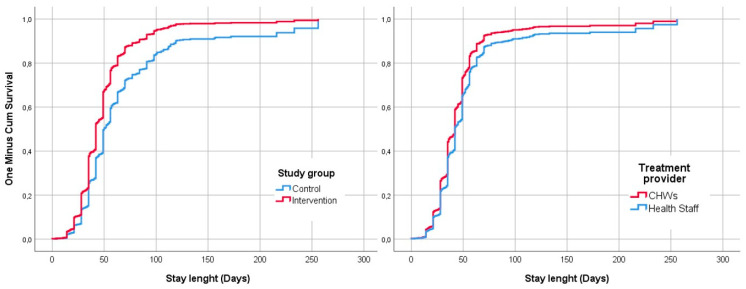
Survival curves for the event recovery adjusted by criteria of admission and compared by study group and treatment provider within the intervention group.

**Table 1 nutrients-13-04067-t001:** Household sociodemographic characteristics compared between the control and intervention communes.

	Control *n* = 223		Intervention *n* = 224		*p* Value
	*n* (mean)	% (SD)	*n* (mean)	% (SD)	
**Demographics**					
Household size	(8.02)	(3.92)	(7.66)	(3.74)	0.321
Female proportion	905	50.2%	865	51.2%	0.555
**Farming as the main source of subsistence**	218	97.7%	217	96.9%	0.602
**House characteristics**					
Cement floor	1	0.1%	1	0.4%	0.526
Palm/leaves roof	118	52.7%	60	26.9%	<0.001
Handmade bricks of earth roof	64	28.6%	111	49.8%	<0.001
Adapted toilet (flush or slab pit)	6	2.7%	9	4.0%	0.445
House in property	218	97.8%	215	96.0%	0.273
**Water source**					
Potable source in the house	75	33.6%	108	41.7%	0.002
Potable source close	4	1.8%	5	2.2%	0.763
Need for disinfection	73	32.7%	76	33.9%	0.250
**Health service for the sick child**					
Health center as the first option for the screening sick child	139	62.1%	128	57.4%	0.312
**Treatment preference**					
Medication of the health center	166	74.4%	144	64.3%	0.021
Traditional self-medication (herbs)	54	24.2%	74	33.0%	
Self-medication with street drugs	6	2.7%	6	2.7%	1.000
Traditional healers	12	5.4%	29	12.9%	0.006
**Barriers to access**					
Fees	82	36.8%	98	42.8%	0.132
Lack of basic infrastructure	52	23.2%	52	23.3%	0.980
Distance	65	29.1%	30	13.4%	<0.001

**Table 2 nutrients-13-04067-t002:** Anthropometric severity at admission compared between study groups.

	Control	Intervention	*p* Value	Effect Size (Cohen’s d)
MUAC Indicators	*n* = 746 Median (IQR)	*n* = 1757 Median (IQR)		
MUAC (mm)	110 (105–113)	112 (110–114)	<0.001	0.324
MUAC z-score	−3.64 (−4.08–−3.20)	−3.40 (−3.80–−3.01)	<0.001	0.335
MUAC quartiles *	% (*n*)	% (*n*)		
Q1 (< 110 mm)	37.4 (279)	19.4 (341)	<0.001	
Q2 (≥ 110 to <111 mm)	20.6 (154)	25.5 (448)	0.009	
Q3 (≥ 111 to <114 mm)	20.1 (150)	25.0 (440)	0.008	
Q4 (≥ 115 mm)	21.8 (163)	30.1 (528)	<0.001	
**WHZ Indicators**	***n* = 760** **Median (IQR)**	***n* = 1894** **Median (IQR)**	***p* Value**	**Effect Size** **(Cohen’s d)**
Weight (kg)	6.0 (5.3–6.8)	6.2 (5.5–7.1)	<0.001	0.203
Height (cm)	68.0 (65.0–73.0)	69.0 (65.0–74.0)	0.008	0.101
WHZ	−3.47 (−4.08–−2.95)	−3.21 (−3.73–−2.56)	<0.001	0.640
WHZ quartiles *	% (*n*)	% (*n*)		
Q1 (< −3.83)	32.6 (248)	21.1 (418)	<0.001	
Q2 (≥ −3.83 to <−3.29)	28.8 (219)	23.5 (446)	0.005	
Q3 (≥ −3.29 to <−2.65)	20.7 (157)	26.7 (505)	0.001	
Q4 (≥ −2.65)	17.9 (136)	27.7 (525)	<0.001	

* Quartile values calculated for the whole sample (control + intervention); IQR: interquartile range; MUAC: middle–upper arm circumference; WHZ: weight for height z-score.

**Table 3 nutrients-13-04067-t003:** Children classified as SAM by WHZ admitted to treatment in a MUAC-only scenario.

	Total SAM by WHZ	Proportion Admitted by MUAC	Proportion Admitted by MUAC with Extended Thresholds		
		<115 mm	<118 mm	<120 mm	<125 mm
**Whole sample**	100% (1513)	81.4% (1231)	90.4% (1368)	92.5% (1399)	99.5% (1506)
**Control**	100% (543)	88.4% (480)	94.8% (515)	96.1 (522)	99.6% (541)
**Intervention**	100% (970)	77.4% (751)	87.9% (853)	90.4% (877)	99.5% (965)
*Health centers*	100% (544)	77.6% (422)	88.8% (483)	91.4% (497)	99.4% (541)
*CHWs*	100% (418)	77.0% (322)	86.8% (363)	89.0% (372)	99.5% (416)

CHWs: community health workers; MUAC: middle upper arm circumference; SAM: severe acute malnutrition; WHZ: weight-for-height Z-score.

**Table 4 nutrients-13-04067-t004:** Treatment outcomes of children with uncomplicated severe acute malnutrition by study group (A) and by treatment provider inside the intervention group (B).

**(A) Whole Sample by Study Group**	**Control** **(*n* = 767)** **% (*n*)**	**Intervention** **(*n* = 1977)** **% (*n*)**	**HR [95% C.I.] ****	***p* Value**
Cured	72.1% (553)	77.2% (1527)	1.613 [1.453–1.789]	<0.001
Default	9.9% (76)	7.7% (153)	1.018 [0.750–1.381]	0.909
Non-respondent	5.9% (45)	7.5% (149)	1.827 [1.269–2.628]	0.001
Medical reference	1.8% (14)	1.4% (27)	0.853 [0.427–1.703]	0.652
Internal transfer	1.0% (8)	2.4% (448)	3.475 [1.442–8.372]	0.006
Death	2.1% (16)	1.8% (38)	0.729 [0.372–1.430]	0.358
Other *	7.2% (55)	1.9% (38)	0.501 [0.323–0.803]	0.002
**(B) Intervention Group by Treatment Provider**	**Healtj staff** **(*n* = 1181)** **% (*n*)**	**CHWs** **(*n* = 782)** **% (*n*)**	**HR [95% C.I.] ****	***p* Value**
Cured	73.1% (863)	83.7% (655)	1.250 [1.118–1.397]	<0.001
Default	8.3% (98)	6.9% (54)	1.036 [0.719–1.495]	0.848
Non-respondent	9.7% (115)	4.3% (34)	0.586 [0.379–0.905]	0.016
Medical reference	1.6% (19)	0.9% (7)	0.686 [0.279–1.702]	0.416
Internal transfer	2.5% (30)	2.0% (16)	0.852 [0.426–1.703]	0.650
Death	2.6% (31)	0.5% (4)	0.231 [0.068–0.785]	0.019
Other *	2.1% (25)	1.7% (13)	0.788 [0.389–1.595]	0.507

* Mostly related to therapeutic food stock-out; ** adjusted by criteria for admission (WHZ, MUAC, both or edema). Reference: (A) control group; (B) health staff. CHWs: community health workers; C.I.: confidence interval; HR: hazard ratio.

**Table 5 nutrients-13-04067-t005:** Recovery time from uncomplicated severe acute malnutrition compared between study groups and by health provider inside the intervention group.

	*n*	Median (Days)	IQR (Days)	*p* Value	Effect Size (Cohen’s d)
**Whole sample**					
Control	553	49.0	35.0–65.0	<0.001	0.582
Intervention	1517	36.0	28.0–49.0		
**Intervention group**					
Health staff	860	42.0	28.0–49.0	0.001	0.170
CHWs	651	35.0	28.0–49.0		
**By admission criteria ***					
				^1 vs. 2^ 0.452	^1 vs. 2^ 0.090
Edema ^1^	58	35.0	21.0–42.5	^1 vs. 3^ 0.004	^1 vs. 3^ 0.214
WHZ only ^2^	217	35.0	18.0–49.0	^1 vs.4^ <0.001	^1 vs. 4^ 0.311
MUAC only ^3^	652	42.0	28.0–49.0	^2 vs. 3^ 0.001	^2 vs. 3^ 0.236
MUAC + WHZ ^4^	858	45.0	34.0–56.0	^2 vs. 4^ <0.001	^2 vs. 4^ 0.416
				^3 vs. 4^ <0.001	^3 vs. 4^ 0.260

Mann–Whitney test; CHWs: community health workers; IQR: interquartile range: MUAC: middle–upper arm circumference; WHZ: weight-for-height Z-score. * *p* value and effect size reported for paired comparisons.

**Table 6 nutrients-13-04067-t006:** Cases of diseases detected and treated in an integrated manner with the severe acute malnutrition.

	Whole Sample				Intervention Group			
	Control	Intervention	RR [95% C.I.]	*p* Value	Health Staff	CHWs	RR [95% C.I.]	*p* Value
	% (*n*)	% (*n*)			% (*n*)	% (*n*)		
**On admission**								
Diarrhea	20.6 (158)	25.7 (518)	1.246 [1.065–1.459]	0.006	25.9 (316)	20.5 (197)	0.965 [0.827–1.125]	0.648
Vomit	13.7 (105)	14.7 (296)	1.072 [0.872–1.317]	0.511	13.0 (158)	17.3 (163)	1.332 [1.079–1.645]	0.008
Fever	3.7 (28)	15.2 (306)	4.155 [2.848–6.063]	<0.001	15.0 (183)	15.4 (121)	1.023 [0.828–1.264]	0.834
Cough	1.7 (13)	13.4 (269)	7.872 [4.540–13.648]	<0.001	13.2 (161)	13.5 (106)	1.022 [0.814–1.284]	0.852
Dehydration	0.3 (2)	0.9 (19)	3.622 [0.846–15.511]	0.083	1.1 (13)	0.8 (6)	0.714 [0.273–1.871]	0.493
Malaria	1.8 (14)	1.4 (28)	0.763 [0.404–1.442]	0.405	1.3 (16)	1.5 (12)	1.161 [0.552–2.441]	0.694
Skin injuries	0.5 (11)	0 (0)	0.114 [0.007–0.133]	0.132	0.7 (9)	0.3 (2)	0.343 [0.074–1.584]	0.170
**During treatment**								
Diarrhea	4.3 (33)	15.1 (305)	3.506 [2.472–4.972]	<0.001	16.5 (202)	12.8 (101)	0.777 [0.622–0.969]	0.025
ARI	1.8 (14)	7.9 (159)	4.308 [2.510–7.393]	<0.001	5.1 (63)	11.8 (93)	2.293 [1.687–3.117]	<0.001
Malaria	5.0 (38)	4.7 (95)	0.948 [0.657–1.369]	0.777	3.3 (40)	7.0 (55)	2.136 [1.435–3.178]	<0.001

ARI: acute respiratory infection; CHW: community health workers; C.I.: confidence interval; RR: risk ratio (reference: control and health staff).

## Data Availability

The data presented in this study are avail-able on request from the corresponding author.

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
