# Peer review of "Impact of Integration of Severe Acute Malnutrition Treatment in Primary Health Care Provided by Community Health Workers in Rural Niger"

_nutrients, 2021, doi:10.3390/nu13114067_

Round 1
Reviewer 1 Report
It is very important and interesting study to guide the policy makers to have informed decision. The authors focused on the Impact of the Integration of Severe Acute Malnutrition Treatment in the Primary Health Care Provided by the Community Health Workers in rural Niger.
Comments
Abstract
Line 25 – 26: Using the abbreviation HFs and sometimes HF without S please check it all over the paper.
Line 32 – 33: The last sentence is not clear for me because there is no any information in the introduction about the adopted policies in Niger, is that mean PHC is not a national policy in the country? Or do you want to say to adopt CHWs in the PHC policy.
Introduction
I suggest adding one paragraph on the health system and adopted policies in Niger if you want to have a recommendation in this regard.
Methods:
The methodology section is not organized and confusing for the reader, I would suggest having structure section with subtitles including; Type of study, Study area include map of the targeted areas in the district, target population and age groups, sample size and sampling method (is sample unit the household or children), analysis, ethical consideration.
It is not clear how many villages, how they selected them, is the cluster include one village, how they selected the cluster, how they selected the household within the cluster and the children in the household,
Add variable conceptual framework to understand the relations and causality in the results later.
Add section to define the anthropometrics measures that used in the study, may add the WHO criteria and Classification of GAM according the WHZ and MUAC in table. What type of measures used – brand name. How did you measure the coverage? What do you mean by Semi-quantitative evaluation?
Were there two surveys baseline and end line, but please explain in detail, if the same methodology followed in both surveys, may describe each survey under subtitle separately to avoid confusion
Line 126-128 about the consent please move it to ethical consideration
Results:
The authors mention in the first paragraph results on socioeconomic but did not show us the results in percentage as it showed in the table, and then linked this part with the prevalence of GAM and SAM is that was part of the pre survey ? is the result here for the end line survey only. Is there difference in prevalence between two surveys or it was not measured in the baseline survey? This should be answered in the methodology.
After Table 1 directly described the children characteristic, I think this could be under the socio economic and may the results reflected clearly in table (Line 179-185).
Table two
The classifications of WHZ and MUAC are confusing for me who I worked in nutrition in the field, what about the readers who have no idea about it. I suggest to used simple classification to ease the understanding use the classification of WHO (SAM and MAM) what the added value to use the four ranges
Figure 2: mentioned Children were distributed according to the anthropometric criteria for admission to treatment by health provider 217 in the intervention study group’’ but in the PIE shows control group not only the intervention group Do you want to compare between the groups but who provide the you with the information in the control group ????
Table 3 title is long need to be rephrased and it was not differentiate between the control and intervention groups.
Table 4: write the HR in full words, or put key word below the table
Discussion
Line 278: Authors did not discuss the reasons that reduced the coverage in the control area. Is there any reasons related to the project itself or some general reasons also could be also in the intervention group?
Line 304: How is it opposite trend? It seems same trend but higher than this study, could you explain?
Line 373: The last sentence as recommendation to conduct more studies, give a negative impression, I would suggest to have more specific recommendations.
Author Response
It is very important and interesting study to guide the policy makers to have informed decision. The authors focused on the Impact of the Integration of Severe Acute Malnutrition Treatment in the Primary Health Care Provided by the Community Health Workers in rural Niger.
Comments
Abstract
Line 25 – 26: Using the abbreviation HFs and sometimes HF without S please check it all over the paper.
Authors: We have made the requested adjustments.
Line 32 – 33: The last sentence is not clear for me because there is no any information in the introduction about the adopted policies in Niger, is that mean PHC is not a national policy in the country? Or do you want to say to adopt CHWs in the PHC policy.
Authors: There is a PHC policy in the country with CHWs, but they are treating just malaria, pneumoniae and diarrhea. Acute malnutrition is not included in the package of activities CHWs have to carry out. At the same time, there is a Community Management of acute malnutrition (CMAM) policy in the country, which dit not allow CHWs to treat acute malnutrition. What we expect from the Ministry of Health is to adapt the management of acute malnutrition policy, allowing CHWs to treat acute malnutrition. We have already modified in the
manuscript
Introduction
I suggest adding one paragraph on the health system and adopted policies in Niger if you want to have a recommendation in this regard.
Authors: Thank you for the suggestion. We have added a paragraph clarifying current policies on the management of malnutrition.
Methods:
The methodology section is not organized and confusing for the reader, I would suggest having structure section with subtitles including; Type of study, Study area include map of the targeted areas in the district, target population and age groups, sample size and sampling method (is sample unit the household or
children), analysis, ethical consideration.
Authors: Thank you for the suggestion. The methodology has been reorganized into subsections and small clarifications have been added. We think that it is now clearer that three assessments have been carried out (socioeconomic, coverage and effectiveness), each with its own specific methodology which implies a different sampling.
At the sub section “study design and location”, we have also moved a paragraph, line 112-117, for a better understanding. Map of the target area. As there is already in the article several tables and figures, we have considered to include it as supplementary material. Line 109.
It is not clear how many villages, how they selected them, is the cluster include one village, how they selected the cluster, how they selected the household within the cluster and the children in the household.
Authors: We have rewritten this part on the methodology of the socioeconomic survey, adding more details to allow the reader to better understand it.
Add variable conceptual framework to understand the relations and causality in the results later.
Authors: We have added in the methodology information on the other variables that have been analyzed together with the discharge outcomes in the effectiveness assessment.
Add section to define the anthropometrics measures that used in the study, may add the WHO criteria and Classification of GAM according the WHZ and MUAC in table.
Authors: Each subsection of the methodology indicates which anthropometric criteria have been considered (for the effectiveness assessment, for the analysis of the prevalence of SAM in the socio-economic description of communes and for coverage assessment). We believe that with the division into subsections, this will be clearer to the reader without the need to add an additional table to the paper.
What type of measures used – brand name. How did you measure the coverage? What do you mean by Semi-quantitative evaluation? Were there two surveys baseline and end line, but please explain in detail, if the same methodology
followed in both surveys, may describe each survey under subtitle separately to avoid confusion
Authors: We have expanded the information on the standardized SQEAC methodology for coverage analysis and have specifically indicated that this same methodology was used at the beginning and end of the study
Line 126-128 about the consent please move it to ethical consideration.
Authors: We have made the requested adjustments.
Results:
The authors mention in the first paragraph results on socioeconomic but did not show us the results in percentage as it showed in the table, and then linked this part with the prevalence of GAM and SAM is that was part of the pre survey?
is the result here for the end line survey only. Is there difference in prevalence between two surveys or it was not measured in the baseline survey? This should be answered in the methodology.
Authors: We have clarified this part in the methodology. On the one hand, the socio-economic study was conducted (with two-stage sampling) before the start of the intervention. Then, once the intervention (treatment of the children) had begun, we returned to those same randomly selected households for the
socio-economic survey, and we measured the MUAC of all children under 5 years of age to calculate the prevalence of GAM and SAM. We believe that having clarified this in the methodology, the results could be now better understood.
After Table 1 directly described the children characteristic, I think this could be under the socio economic and may the results reflected clearly in table (Line 179-185).
Authors: We would like to clarify this point. Paragraph after table 1, is explaining those children who have been including in the study, during the research. Information in table 1, is coming from the socio-economic survey. We have modified some sentences and together with the restructuring of the methodology, we believe that this is now clearer.
Table two
The classifications of WHZ and MUAC are confusing for me who I worked in nutrition in the field, what about the readers who have no idea about it. I suggest to used simple classification to ease the understanding use the classification of WHO (SAM and MAM) what the added value to use the four ranges
Authors. In the methodology, we have explained in detail the rationale for this analysis. Reporting the % of MAS cases by WHO standard indicators would not allow us to compare severity at admission. We want to know the most severe cases (anthropometries even less than 115mm or a WHZ <-3). Since the WHO does not indicate any specific cut-off point to differentiate the most severe cases, an objective way to do this is to divide the sample into quartiles and consider those that fall in Q1 as the most severe cases. We have added this clarification also in the methodology.
Figure 2: mentioned Children were distributed according to the anthropometric criteria for admission to treatment by health provider 217 in the intervention study group’’ but in the PIE shows control group not only the intervention group Do you want to compare between the groups but who provide the you with the
information in the control group
Authors: Yes thank you very much, there was a mistake, we want to compare it between study groups. Data within the intervention group are given directly in the text (not included in the figure). We have changed the title of Figure 2 to avoid confusion.
Table 3 title is long need to be rephrased and it was not differentiate between the control and intervention groups.
Authors: We have reduced the table title and rephrase column headings also. Results are shown for the total sample, for each study group and, within the intervention group, for each treatment provider (HFs and CHWs). We believe that comparisons between groups are not necessary here since we do not expect it to vary between groups. This is provided to show that by increasing the
MUAC threshold, the majority of children who have SAM by WHZ could be included into treatment. This is a result that we have been specifically requested by health authorities given the debate about the possibility of using MUAC-only protocols in low-resource or low-skilled settings (such as CHWs).
Table 4: write the HR in full words, or put key word below the table.
Authors: We have made the requested adjustments.
Discussion
Line 278: Authors did not discuss the reasons that reduced the coverage in the control area. Is there any reasons related to the project itself or some general reasons also could be also in the intervention group?
Authors: We have made the requested adjustments.
Line 304: How is it opposite trend? It seems same trend but higher than this study, could you explain?
Authors: Yes, we have made the requested adjustments.” However, studies using national cross-sectional data show different results, and more children were identified by WHZ (51% vs. 24%)”
Line 373: The last sentence as recommendation to conduct more studies, give a negative impression, I would suggest to have more specific recommendations.
Authors: Yes, we have made the requested adjustments “Further analysis would be needed to identify how to make the work of CHWs operational on a large scale in the country, and also whether the same good results in terms of effectiveness and coverage would be achieved if CHWs work in a humanitarian context”
Reviewer 2 Report
The authors present an overall well-prepared manuscript. However, I do have some minor comments for improvement:
- Title: I suggest to delete most “the” in the title: “Impact of Integrating Severe Acute Malnutrition Treatment in Primary Care Provided by Community Health Workers in Rural Niger”
- Line 22: The acronym needs to be introduced.
- Line 40: It has to be squared.
- Line 44: Reference missing.
- Lines 67 and 69: I guess it should be “district” instead of “department”.
- Line 114: Please elaborate the assumptions for the sample size estimation here and not in lines 149ff.
- Line 134: How can the recruitment be conducted until March 2019 when this equals the end-line questionnaire?
- Lines 173-175: Please delete, because this is part of the journal’s instructions.
- Figure 1 is not readable in black/white format.
- Figure 2 is not needed. It can be deleted, when the results are described in the main text.
- Table 4: Please check the CI for the HR of “Other”. There seems to be a mistake.
Author Response
The authors present an overall well-prepared manuscript.
However, I do have some minor comments for improvement:
Title: I suggest to delete most “the” in the title: “Impact of Integrating Severe Acute Malnutrition Treatment in Primary Care Provided by Community Health Workers in Rural Niger”
Authors: Thank you for your this suggestion. We have made the requested adjustments
Line 22: The acronym needs to be introduced.
Authors: We have made the requested adjustments, “severe acute malnutrition “
Line 40: It has to be squared.
Authors: We have made the requested adjustments, KM2
Line 44: Reference missing.
Authors: There were a mistake, we have made the requested adjustments in reference 1
Lines 67 and 69: I guess it should be “district” instead of “department”.
Authors: We have made the requested adjustments
Line 114: Please elaborate the assumptions for the sample size estimation here and not in lines 149ff.
Authors: These are different assessments, each with its own methodology and independent sample calculation according to the standardized methodology chosen. We have restructured the methodology section, adding more details to each section so that we believe that now it is better understood and these confusions are avoided.
Line 134: How can the recruitment be conducted until March 2019 when this equals the end-line questionnaire?
Authors: As mentioned above, we have three different assessments. On the one hand, the socioeconomic survey was conducted before starting the rest of the assessments (March 2018).
Then the initial coverage survey (April 2018) was conducted before starting the intervention, in order to reflect the pre-intervention situation. After that, the intervention itself, to which inclusion of children in treatment starts in June 2018 and continued until March 2019, when the final coverage survey was conducted (March 2019). We think that with the new methodology subsections and clarifications it would be clearer for the readers now.
Lines 173-175: Please delete, because this is part of the journal’s instructions.
Authors: Thank you very much. We have made the requested adjustments
Figure 1 is not readable in black/white format.
Authors: We have elaborated the figures in color following the recommendations given by the journal in the instructions for authors where it expressly states the following: "Authors are encouraged to prepare figures and schemes in color (RGB at 8-bit per channel). There is no additional cost for publishing full color graphics".
Figure 2 is not needed. It can be deleted, when the results are described in the main text.
Authors. The results shown in Figure 2 are not described in the text. In the text we only provide the p-value of the comparison but do not give the percentages of each type of admission in each group, that can only be seen in the figure. We believe it is easier and more visual for readers to give the percentages in the
form of a pie chart than to put the value of the six percentages in the text. What we do expand on in the text is the value of the percentages comparing between treatment providers within the intervention group
(CHWs vs. HFs).
Table 4: Please check the CI for the HR of “Other”. There seems to be a mistake.
Authors: Yes thank you very much. We have made the requested adjustments
Round 2
Reviewer 1 Report
Thank you for the authors, the manuscript is improved in term of clarity, I would suggest to explain more the admission criteria based on the severity of malnutrition in the methodology may add small table or in text that will make it clear and to be used by other studies in the future.
Author Response
REVIEWER COMMENT
Thank you for the authors, the manuscript is improved in term of clarity, I would suggest to explain more the admission criteria based on the severity of malnutrition in the methodology may add small table or in text that will make it clear and to be used by other studies in the future.
AUTHORS ANSWER
Dear Reviewer,
We sincerely thank you for taking the time to do this second review and for your new comment, which allows us to improve our work for a better understanding of future readers.
As we mentioned to the other reviewer in the first phase of the review, there are no formal definitions of what is considered a case of extreme severity in the context of community-based management of acute malnutrition.
The UN agencies (WHO, UNICEF) do not incorporate this concept in their international recommendations, only the cut-off point at which a child should be treated with therapeutic feeding (not just supplementation), which is WHZ < -3 z-score or MUAC < 115. But they do not indicate what values below these would be considered extreme severity enough to take other complementary actions (e.g., hospital admission instead of community management). That is why we decided to apply objective (and replicable) criteria, such as dividing the anthropometric measurements recorded at admission into quartiles based on the median of all the children included in the study.
However, we understand that the same doubt that has arisen for you may also arise for readers. Therefore, we have incorporated this explanation within the manuscript in the methodology section (marked in yellow within the document).
Thank you again for your time and dedication.
Best regards,